# An novel cloud task scheduling framework using hierarchical deep reinforcement learning for cloud computing

Delong Cui[1]*, Zhiping Peng[2], Kaibin Li[1], Qirui Li[1], Jieguang He[1], Xiangwu Deng[1]

**1** College of Electronic Information Engineering, Guangdong University of Petrochemical Technology, Maoming, China, **2** Jiangmen Polytechnic, Jiangmen, China

* delongcui@gdupt.edu.cn

## Abstract

With the increasing popularity of cloud computing services, their large and dynamic load characteristics have rendered task scheduling an NP-complete problem. To address the problem of large-scale task scheduling in a cloud computing environment, this paper proposes a novel cloud task scheduling framework using hierarchical deep reinforcement learning (DRL) to address the challenges of large-scale task scheduling in cloud computing. The framework defines a set of virtual machines (VMs) as a VM cluster and employs hierarchical scheduling to allocate tasks first to the cluster and then to individual VMs. The scheduler, designed using DRL, adapts to dynamic changes in the cloud environments by continuously learning and updating network parameters. Experiments demonstrate that it skillfully balances cost and performance. In low-load situations, costs are reduced by using low-cost nodes within the Service Level Agreement (SLA) range; in high-load situations, resource utilization is improved through load balancing. Compared with classical heuristic algorithms, it effectively optimizes load balancing, cost, and overdue time, achieving a 10% overall improvement. The experimental results demonstrate that this approach effectively balances cost and performance, optimizing objectives such as load balance, cost, and overdue time. One potential shortcoming of the proposed hierarchical deep reinforcement learning (DRL) framework for cloud task scheduling is its complexity and computational overhead. Implementing and maintaining a DRL-based scheduler requires significant computational resources and expertise in machine learning. There are still shortcomings in the method used in this study. First, the continuous learning and updating of network parameters might introduce latency, which could impact real-time task scheduling efficiency. Furthermore, the framework's performance heavily depends on the quality and quantity of training data, which might be challenging to obtain and maintain in a dynamic cloud environment.

**Data availability statement:** All relevant data are within the paper.

**Funding:** Key Realm R&D Pro-gram of Guangdong Province(2021B0707010003); National Natural Science Foundation of China (62273109); Guangdong Basic and Applied Basic Research Foundation (2022A1515012022, 2023A1515240020, 2023A1515011913); Key Field Special Project of Department of Education of Guangdong Province (2024ZDZX1034); Maoming Science and Technology Project (210429094551175, 2022DZXHT028, mmkj2020033); Projects of PhDs' Start-up Research of GDUPT (2023bsqd1002, 2023bsqd1013, XJ2022000301); Special Innovation Projects for Ordinary Universities in Guangdong Province in 2023 (2023KTSCX086). The funders had no role in study design, data collection and analysis, decision to publish, or preparation of the manuscript.

**Competing interests:** The authors have declared that no competing interests exist.

## 1. Introduction

Cloud computing is a resource delivery and usage model. Service providers integrate many nodes into a unified resource pool through virtualization technology, and users obtain the required computing resources through the network [1]. Cloud computing, as one of the core infrastructures in the current field of information technology, faces increasing pressure in task scheduling with the rapid development of big data, the Internet of Things, and 5G technologies. Task scheduling, whose essence is to reasonably allocate user requests to computing nodes for processing, is an important research direction in cloud computing. However, this generates a large solution space, and the optimal solution cannot be obtained in polynomial time, thus, the task scheduling of cloud computing is an uncertain NP problem [2,3].

Traditional task scheduling methods, whether heuristic algorithms [4] based on simple rules or some metaheuristic algorithms [5], exhibit numerous limitations when dealing with large-scale and dynamically changing cloud tasks [6]. For example, heuristic algorithms often lack adaptability to complex environmental changes and have difficulty flexibly adjusting scheduling strategies under different load and resource conditions. Although metaheuristic algorithms can perform global optimization to a certain extent, they have complex parameter settings and high computational overhead, making them difficult to apply effectively in cloud task scheduling scenarios with high real-time requirements.

Many researchers have studied this problem and proposed heuristic and metaheuristic algorithms to solve it. However the actual cloud computing environment is complicated and dynamic, and traditional methods cope poorly with it. Researchers use reinforcement learning (RL) and deep reinforcement learning (DRL) for learning capabilities to solve the dynamic scheduling problem of cloud computing [7–9]. Owing the diversity of user requests and resources, different quality of service (QoS) constraints must be met simultaneously, and determining how to respond to large-scale user requests while meeting the requirements of cloud service providers is an urgent problem. Intelligent scheduling algorithms are essential for overcoming the difficulties of large-scale task scheduling. In this research, a hierarchical intelligent task scheduling framework (HITS) based on a hierarchical DRL algorithm is proposed. In the scheduling framework, a collection of virtual machines (VMs) is called a VM cluster. When the framework receives a task request, it allocates the task to a cluster, and then to a VM via the task scheduler inside the cluster. We apply DRL technology to the scheduler, and through the design of the state space and return function of each layer, it can adapt to the dynamic changes in the cloud computing environment, and adjust its scheduling strategy through continuous learning.

On the basis of these current situations, we propose adopting hierarchical DRL technology to address the cloud task scheduling problem. DRL has powerful learning capabilities and adaptability to complex environments. It can automatically optimize scheduling strategies through continuous interaction and learning with the cloud environment. The hierarchical architecture helps to decompose large-scale problems into manageable subproblems, improving decision-making efficiency and system scalability. We expect that through this innovative method, it is possible to meet the cost

control requirements of cloud service providers while providing users with more efficient and reliable services, achieving a comprehensive improvement in multiple aspects such as performance, cost, and flexibility in cloud computing task scheduling, filling the gaps of traditional methods in handling large-scale and dynamic cloud task scheduling, and promoting the further development and application of cloud computing technology in modern information technology systems.

Cost and load balancing are two crucial objectives in cloud task scheduling. From the perspective of cost, cloud service providers need to reduce the cost of resource usage as much as possible to enhance profit margins while meeting user requirements. Different types of sVMs have diverse cost structures, including computing costs, storage costs, and bandwidth costs. Our scheduler, through the DRL algorithm, comprehensively considers the resource requirements of tasks and the cost characteristics of VMs during the task allocation process. For example, when a task arrives, the scheduler evaluates the idle resource situation and the corresponding cost of the VMs within each cluster and preferentially assigns the task to the combination of VMs or clusters that can meet the task requirements and have a lower cost. This approach can effectively reduce the overall cost of task execution and improve the cost-effectiveness of resources.

For load balancing, the motivation is to ensure that the utilization rates of various resource nodes (clusters and VMs) in the cloud environment are relatively balanced and avoid situations where some nodes are overloaded while others are idle. This not only helps to improve the overall performance and stability of the system but also extends the service life of hardware devices. In the decision-making process, our scheduler takes the load situations of clusters and VMs as important state information and inputs it into the DRL model. By designing a reasonable return function, positive rewards are given to scheduling decisions that can achieve load balancing, and vice versa. For example, when the standard deviation of the virtual machine loads within a cluster is small, indicating a relatively balanced load, the scheduler tends to continue assigning tasks to this cluster. When the load of a certain virtual machine is too high, the scheduler will consider assigning subsequent tasks to other VMs or clusters with lighter loads, thereby dynamically adjusting the task allocation strategy to achieve load balancing of resources in the cloud environment and reducing performance bottlenecks and resource waste caused by uneven loads.

This research proposes an innovative hierarchical intelligent task scheduling (HITS) framework based on the hierarchical DRL algorithm to address the challenge of large-scale task scheduling in cloud computing. Compared with traditional methods, HITS has significant advantages. First, through hierarchical partitioning and effective manipulation of the solution space, it accelerates the task scheduling process and simultaneously optimizes the task overdue time and cost, which is particularly crucial in large-scale task scheduling scenarios. Second, the model structure and return function of DRL are meticulously designed in accordance with the dynamic characteristics in the cloud environment. In response to the dynamic variation in the number of VMs, by modeling the Gaussian distribution of relevant features and using it as state information, the model can adaptively adjust. For different load conditions, a unique reward function is designed, which feeds back rewards on the basis of the load to drive the model to learn corresponding decision-making strategies, thereby achieving efficient and intelligent task scheduling in a complex and variable cloud environment.

The structure of this article is organized as follows: Chapter 2 reviews research related to cloud computing scheduling. Chapter 3 describes the key stages and problem models within a large-scale task scheduling framework. Chapter 4 provides a detailed account of the core design of the algorithm used in this project, including its fundamental principles, the design of the main state space and reward functions, and the underlying design concepts. Chapter 5 outlines the process of evaluating the algorithm's performance through simulation experiments. Finally, Chapter 6 summarizes our findings and discusses potential future research directions.

## 2. Related work

Task scheduling is a challenging problem in cloud computing; an efficient, intelligent, minimum-cost strategy is a key factor affecting performance. Researchers have proposed various heuristic and metaheuristic task scheduling algorithms, and DL and DRL algorithms are emerging.

The most common task scheduling algorithms have been heuristic. However, the heuristic approach prioritizes versatility, which is easy to understand and implement, but cannot adopt a more ideal strategy according to the environment. The metaheuristic approach requires appropriate parameter settings to escape from local optimal solutions and accelerate convergence. Common heuristic scheduling strategies include the Min-Min and Max-Min algorithms [10]. Although the traditional Min-Min approach can significantly reduce the total task scheduling time, its tendency to prioritize high-performance nodes leads to poor load balancing and higher costs under low load conditions. In contrast, Max-Min achieves more balanced scheduling by prioritizing larger tasks, but this can result in increased overall response time due to potential delays in executing smaller tasks. The round robin method is a simpler scheduling strategy that assigns tasks sequentially to each computing node to achieve balance. However, widely differing calculation times of tasks or processing capacities of computing nodes can lead to an unbalanced load and reduced resource utilization [11].

Metaheuristic algorithms minimize the execution cost of task schedules by searching for the optimal solution while meeting deadlines, and are often used in workflow task scheduling. Soulegan et al. [12] proposed a multipurpose weighted genetic algorithm to minimize the time and cost of cloud task scheduling. The algorithm includes several parameters for comprehensive optimization, such as utility, task execution cost, response delay, waiting time, total completion time, and throughput. Its goal is to address the task scheduling challenges in cloud computing, Ababneh et al. [13] proposed a composite multiobjective strategy—hybrid gray wolf and whale optimization (HGWWO)—which integrates the gray wolf optimizer (GWO) and the Whale Optimization Algorithm (WOA) to reduce costs, energy consumption, and overall execution time, while also increasing resource utilization efficiency. A delay-aware scheduling algorithm based on VM matching and employing metaheuristic methods was proposed to optimize resource allocation and reduce the task response time [14]. This approach is built on tabu search and enhanced by incorporating approximate nearest neighbor (ANN) and fruit fly optimization (FOA). To improve task execution efficiency, quality of service (QoS), and energy management, the EHEFT-R scheduling algorithm was introduced [15]. Additionally, a multiobjective simulated annealing (MOSA) algorithm was developed for effectively allocating tasks on fog and cloud nodes while meeting deadline constraints [16]. A goal planning method (GPA) was used to identify the optimal solution among the nondominated solutions for multiple objectives. Nakrani et al. [17] proposed a load balancing task scheduling approach based on genetic algorithms, the experimental results of which were shown to be effective in reducing the overall delay and energy consumption. Jia et al. [18] developed a dual-objective task scheduling model that integrates a queuing model for delay estimation and an energy consumption model for heterogeneous resources. Additionally, a large-scale task scheduling framework based on Pareto optimization was introduced to schedule large tasks within a time unit. For the multiobjective scheduling problem, an elite learning Harris hawk optimization algorithm (ELHHO) was proposed to enhance the exploration capability of the standard HHO algorithm through elite adversarial learning techniques [19]. A task scheduling algorithm based on the K-medoid particle swarm (KMPS) method was proposed to reduce the task completion time and lower the maximum completion time by introducing a weighting mechanism [20]. To decrease the turnaround time and enhance resource utilization, a scheduling strategy based on particle swarm optimization (PSO) was introduced, aiming to allocate applications efficiently to cloud resources while considering transmission costs and the current load [21]. This strategy incorporates inertia weights to avoid local optima. Despite the significant contributions of these studies, there are still limitations in problem modeling and parameter tuning of heuristic algorithms.

Reinforcement learning (RL) is utilized to address task scheduling issues because a single scheduling algorithm cannot optimize scheduling performance in real-time control systems within complex online task scheduling environments. An optimization framework based on Q-learning was proposed to select an appropriate scheduling algorithm for a mixed task set [22]. Through continuous learning, the scheduler can adapt to a complex dynamic environment. A low-load task scheduling approach using Q-learning dynamically adapts the scheduling strategy in response to network changes in the edge computing environment, aiming to keep the overall load low while balancing the load and packet loss rates [23]. An energy-saving cloud computing task scheduling framework (QEEC) based on Q-learning was proposed to save energy

while meeting user needs [24]. A centralized task scheduling system employs the M/M/S queuing model in the initial phase to allocate user requests to various servers in the cloud. Each server ranks requests on the basis of task looseness and lifecycle, whereas a Q-learning-based scheduler assigns tasks to VMs. A meta-reinforcement learning-based offloading method achieves rapid adaptation to new environments through a limited number of gradient updates and samples [25]. The mobile application is represented as a directed acyclic graph (DAG), and the offloading strategy is modeled using a sequence-to-sequence (seq2seq) neural network. A method that combines a first-order approximation and a tailored agent target accelerates network training. Traditional reinforcement learning (RL) methods are limited by their reliance on discrete state representations and sensitivity to parameter tuning. They often require significant computational resources and training data to achieve satisfactory performance in complex environments. In contrast, DRL addresses these limitations by leveraging neural networks to handle large and continuous state spaces, improving convergence speed, and providing more flexible and automatic parameter tuning.

RL usually uses Q tables to record the mapping of state and expected returns. This approach is effective for small-scale problems, but is inapplicable when the states are large in number or continuous. DRL solves this problem by fitting the relationship between states and re-turns through the powerful function fitting of deep neural networks. These solutions are also being applied to task scheduling. Multiple tasks are scheduled for a VM configured on an edge server to maximize long-term task satisfaction [26]. A strategy-based REIN-FORCE algorithm was proposed, and a fully connected neural network (FCN) was used as a decision model. The computational cost under resource and deadline constraints was minimized and a tailored dual deep Q-learning algorithm (CDDQL) using a target network and empirical relay technology was proposed [27]. A DRL model focused on QoS feature learning for optimizing energy consumption and QoS was proposed [28]. This approach uses an enhanced stacked denoising autoencoder to extract more reliable QoS feature information. For decision-making, a cooperative resource scheduling algorithm based on reinforcement learning (RL) for multipower machines was introduced, achieving an effective balance between energy savings and QoS enhancement. A DRL-based solution was proposed to address cloud resource management problems [29], which uses convolutional neural networks to extract the characteristics of resource management models, and simulation learning to accelerate model training, and demonstrated the potential of integrating imitation learning and DRL to solve hard resource scheduling problems.

Although certain progress has been achieved in these studies, some gaps or limitations still exist, especially in terms of insufficient adaptability to dynamic environments and limited capabilities in handling complex resource management issues. In view of the abovementioned challenges and limitations, this research proposes a hierarchical intelligent task scheduling (HITS) framework based on the hierarchical deep reinforcement learning algorithm. The complexity and dynamism of the cloud computing environment, as well as the need to efficiently handle large-scale tasks while satisfying diverse quality of service (QoS) constraints, constitute the foundation of this research. In the proposed HITS framework, a group of VMs is defined as a VM cluster. When a task request arrives, it is first allocated to a cluster and then assigned to a specific VM through the task scheduler within the cluster. By applying deep reinforcement learning technology to the scheduler and meticulously designing the state space and re-turn function of each layer, this framework can adapt to the dynamic changes in the cloud computing environment and continuously adjust its scheduling strategy through learning and network parameter updates.

## 3. Materials and methods

Advantages of the HierarchicalDRL Technology Employed in This Research over Existing Works:

Advantages of the Hierarchical Architecture: Compared with traditional single-layer task scheduling methods, the hierarchical architecture of HITS can effectively reduce the complexity of the problem. By dividing the task scheduling process into two levels, namely the cluster level and the virtual machine level, the decision space at each level is decreased, and the scheduling efficiency is increased. In cluster-level scheduling, clusters suitable for task processing can be rapidly screened out, avoiding a global search among all VMs, and thereby significantly shortening the task allocation time. This

hierarchical approach is also conducive to resource management and optimization, better balancing the loads among different clusters and VMs and improving resource utilization.

Adaptability of the DRL Model: The DRL model in this research, through a meticulously designed state space and return function, demonstrates remarkable adaptability to tdynamic changes in the cloud environment. Unlike traditional rule-based or heuristic scheduling algorithms, the DRL model can automatically learn and adapt to the dynamic changes of tasks and resources in the cloud environment. For instance, by modeling the Gaussian distribution of changes in the number of VMs, the model can promptly increase or decrease the number of virtual machine resources and adjust the task allocation strategy accordingly. When confronted with different load situations, the unique reward function can guide the model to make a reasonable trade-off between task overdue time and cost, thereby achieving satisfactory scheduling performance under various complex load conditions.

Learning and Optimization Capabilities: The DRL model possesses powerful learning and optimization capabilities. Compared with traditional static scheduling algorithms, it can continuously learn in the process of ongoing task scheduling and constantly optimize its own scheduling strategy. Through techniques such as experience replay and target networks, the model can effectively utilize historical empirical data for learning, avoid becoming trapped in local optimal solutions, and gradually converge to a better scheduling strategy. Such learning and optimization capabilities enable the HITS framework to adapt continuously to changes in the cloud environment and continuously improve the efficiency and quality of task scheduling during long-term operation.

## 3.1. Deep Q-Learning technique

The problem in RL addresses how an agent can maximize its rewards in a complex, uncertain environment. Fig 1a shows a schematic diagram of the agent and environment, which always interact in the RL process. The agent obtains the state $s_t$ in the environment, and the agent uses this state to output an action according to strategy $\pi$. This decision is put into the environment, which outputs the next state $s_{t+1}$, and the reward $r_t$ obtained by the current decision. The reward is a scalar feedback signal given by the environment, which shows how well the agent has adopted a strategy at a certain step. RL maximizes the agent's reward. The agent seeks to maximize its expected cumulative reward. Therefore, we can use RL to acquire a target strategy in an uncertain environment through the design of the state, agent, and reward function [30].

The traditional RL method designs a Q-table to store the updated state action value function until it converges to the optimal strategy Q* suitable for low dimensional discrete scenarios. However, in reality, most scenarios are continuous, and applying RL algorithms can easily lead to the problem of dimensionality disaster. To address this issue, the Deep-Mind team proposed an innovative deep Q-network (DQN) algorithm [31] that combines the advantages of deep neural networks and reinforcement learning to address the challenges faced by traditional methods in high-dimensional spaces. The main idea of this algorithm is to use neural networks to approximate the optimal value function to solve the problem of high-dimensional state features. Usually, this network is referred to as, where are the parameters of the neural network. Initially, they are randomly initialized and then continuously learned on the based of the experience of the agent until the predicted output values of the DQN can be as close as possible to Q*(S, A) for all inputs S and A. Fig 1b shows the neural network structure of a typical DQN algorithm, where the input of the model is the current state of the system environment, and the output is the Q value of each action in the action space. After the model outputs, the agent selects the action with the highest Q value to execute according to the learning criteria.

The (DQN used in this study refers to the Q-learning algorithm based on DL, which combines value function approximation and neural network technology, and uses the target network and experience playback for network training [32].

In the DQN algorithm, the most important consideration is solving for the parameter $\theta$. Typically, DQN models are trained using the time difference algorithm (TD), whose core is to update old estimates with partially fact based estimates (adjusting the parameter ($\theta$). In other words, the loss function of typical deep learning algorithms is calculated on the basis

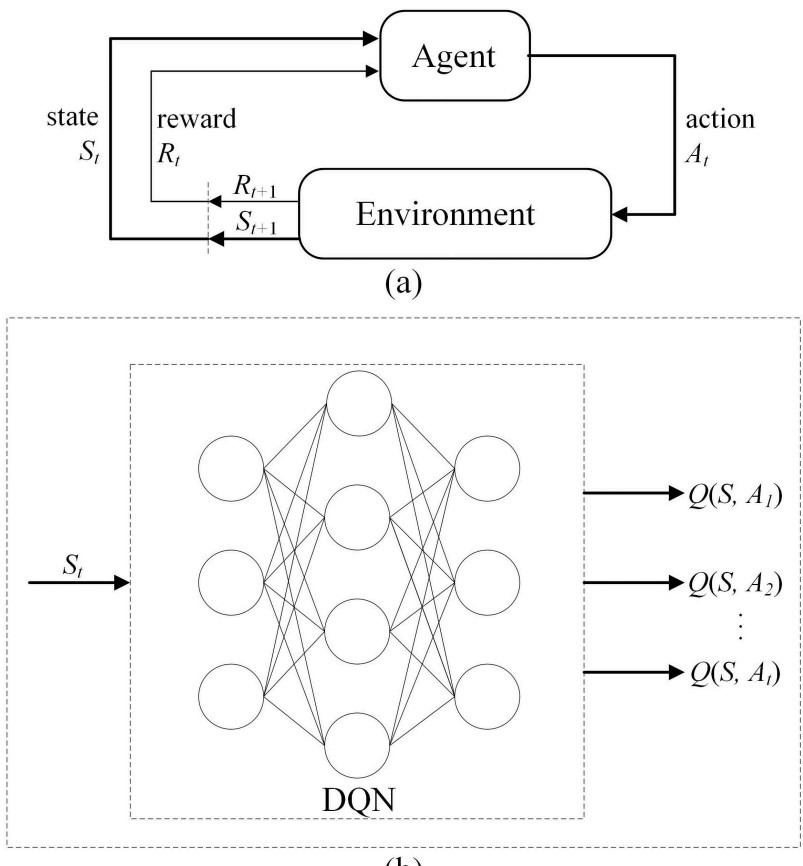

**Fig 1. Reinforcement learning paradigm.** (a) schematic diagram of the agent and environment; **(b)** DQN neural network model mechanism.

of the functional relationship between the predicted data and labeled data, whereas DQN algorithms can incorporate historical empirical data as part of labeled data. Therefore, the first step in training is to calculate the target value via the Bellman equation:

$$Q_{t\,arg\,et}=R_t + \lambda \max_{A_{t+1}} Q(S_{t+1}, A_{t+1}, \theta) \tag{1}$$

Among them, $S_{t+1}$ represents the next state of the environment, $A_{t+1}$ represents the next action to be executed by the agent, $R_t$ is the reward feedback from the environment after the current action is executed, and $Q_{target}$ represents the target value to be fitted by the neural network. The second step calculates the squared difference between it and the predicted value output by the neural network to define the loss function, and further solves for the parameter $\theta$, which can be expressed as follows:

$$L(\theta)=E\left[(R + \lambda \max_{A_{t+1}} Q(S_{t+1}, A_{t+1}, \theta) - Q(S_t, A_t, \theta))^2\right] \tag{2}$$

Among them, $Q(S_t, A_t, \theta)$ is the output of the current neural network, denoted as $Q_{prediction}$. In the third step, the gradient of the loss function is calculated as follows:

$$\frac{\partial L(\theta)}{\partial(\theta)} = E\left[R + \lambda \max_{A_{t+1}} Q(S_{t+1}, A_{t+1}, \theta) - Q(S_t, A_t, \theta)\frac{\partial Q(S_t, A_t, \theta)}{\partial(\theta)}\right] \tag{3}$$

Finally, gradient descent updates the parameter $\theta$:

$$\theta \leftarrow \theta + \alpha\frac{\partial L(\theta)}{\partial(\theta)} \tag{4}$$

where $\alpha$ is the learning rate. By following these four steps, one update of the neural network parameters is completed. The key to training the DQN algorithm lies in updating the target value with historical empirical data while performing gradient descent. This fact based estimation improves the learning performance of the network, allowing it to approach the optimal value function and complete the agent's learning.

In particular, $Q$ is updated based on the error between its prediction and the actual value. The discount factor $\lambda$ is used to calculate the cumulative return. The error calculation adopts the time difference method to reduce the error between the sum of the current return and the cumulative return of the next state gradually. $Q$ is trained through the direct prediction of the cumulative return of the current state, and the loss is multiplied by the learning factor $\alpha$ to update the prediction value.

We note the following:

(1) Epsilon Greedy ($\epsilon$-greedy): $\epsilon$ is a value greater than 0 and less than 1. During the exploration process, a probability equal to $1 - \epsilon$ determines the action according to the Q-function, and can increase the agent's exploration ability. This parameter gradually decreases in the later stage of training to hasten convergence.

(2) Target Network: The constant updating of Q causes training to become unstable. A copy of the original Q network, which is called the target network, is used to predict the next state, and it is copied again after several updates of Q.

(3) Experience Replay: Storing past interaction records and using them in subsequent training can effectively improve the experience utilization rate, and random sampling from experience during training can increase the diversity of training data.

## 3.2. System model

Our task scheduling system is based on a common cloud computing scenario, PaaS, which includes the user who submits the task, the service provider, and the infrastructure as a service (IaaS) provider. Service providers rent VM instances from IaaS providers to build their own resource pools and provide users with various services through the network. There are two main links in this process. Users submit requests, which the service provider dispatches to various computing nodes, and the service provider dynamically leases resources from the IaaS provider on the basis of load pressure. These two links have a common goal of balancing the profit of the service provider and the user experience (such as response time and throughput). This studystudies the task scheduling link.

In actual cloud computing scenarios, there are often large numbers of user requests and computing nodes, and tasks are allocated to nodes through a preset strategy. Therefore, the impact of the task scheduling strategy on the task completion time and execution cost cannot be ignored. A large number of tasks and nodes create a very large solution space. To quickly find a suitable scheduling strategy in this space, we propose a DRL-based hierarchical scheduling framework. First, at each decision point of task scheduling, the scheduler, as the agent in reinforcement learning, interacts with the cloud environment. The agent acquires the state information of the cloud environment, which includes task attributes (such as resource requirements, expected completion time, etc.), cluster resource status (such as computing power, bandwidth, number of idle VMs), and detailed information of VMs (such as current load, processing speed, and

cost). Through the integration and processing of this multidimensional information, a state space is constructed that can comprehensively reflect the matching relationships between tasks and resources as well as the dynamic changes of the cloud environment. On the basis of this state space, the agent outputs an action according to the policy network in the DRL model, and this action is the decision to allocate the task to a specific cluster or virtual machine. After this decision is executed in the cloud environment, the environment feeds back a reward signal to the agent. This reward signal is calculated on the basis of our meticulously designed return function, which comprehensively considers multiple objectives such as task overdue time, cost, and load balancing. If a scheduling decision can reduce task overdue time, lower cost, and contribute to load balancing, the agent will receive a relatively high reward; otherwise, it will receive a lower reward or even a penalty. By continuously repeating this interaction process between the agent and the environment, the DRL model utilizes the experience replay mechanism to store historical interaction data and updates the model parameters on the basis of the temporal difference method, enabling the model to gradually learn the optimal task scheduling strategy under different cloud environment states. In this way, DRL technology is deeply integrated into the entire task scheduling process, from state perception, and decision-making to strategy optimization, realizing the intelligent and dynamic management of cloud task scheduling.

Fig 2 shows the two-layer structure in our proposed scheduling framework, in which we classify several VMs into a virtual machine cluster. As shown in Fig 2a, a task will reach the first-level scheduler after submission, and will be scheduled for a virtual machine cluster. As shown in Fig 2b, within each virtual machine cluster, a secondary scheduler schedules tasks to a VM task processing queue. A VM has its own user waiting queue, according to which it executes tasks in a first-come, first-served manner. In this study, the submission of tasks is dynamic, the number of tasks and resource requirements at a given moment are random, and the task arrival time conforms to a Poisson distribution. We design a scheduling framework to optimize the makespan, cost, and task response time, so that it can adapt to dynamic load and environment changes and obtain the current scheduling strategy through continuous learning.

In our problem model, the resource requirements and the number of tasks submitted by users are not limited, and multiple task types are allowed. Therefore, task-related information is not predictable, and can be obtained only when the task arrives at the scheduler. Service providers can dynamically adjust the number and types of VMs on the basis of load conditions, because our task scheduling framework also considers how to adapt to changes in the number of VMs. Furthermore, we stipulate that the scheduler can obtain the status information of all VMs, including the task processing queue. After execution, task-related information is recorded in the log, including the task assigned to a VM, its start and end times, and its execution cost. This information is used for model training in the scheduling system after processing.

### 3.3. Problem formulation

The experiments in this study were carried out in a simplified simulation environment. We abstracted a task into attribute characteristics, such as millions of instructions per second (MIPS), bandwidth, task instruction volume, bandwidth transmission value, and task overdue completion time. The VM was abstracted into instruction execution speed, bandwidth, and cost. We ignore unexpected failures that may occur in a real environment. In our setting, any number of tasks could arrive simultaneously. The time interval conformed to a Poisson distribution with a certain arrival rate, and the number of tasks and resource requirements were within certain limits. VMs perform tasks on a first-come, first-served basis. The scheduling framework divides the task scheduling process into three parts: the user submits the task to the first-level scheduling center, which schedules the task to a virtual machine cluster, and the second-level scheduler schedules the task to a VM. We model these three parts as follows. In the user submission phase, several tasks arrive at the same time and are stored in the task buffer queue. The first-level scheduler extracts tasks from the task buffer queue for scheduling. A task is defined as

$$t_i = \{ \begin{array}{c} t_i^{mips}, t_i^{bw}, t_i^{mips-l}, t_i^{bw-l}, t_i^{dead} \\ t_i^{load}, t_i^{finish}, t_i^{start}, t_i^{cost} \end{array} \} \tag{5}$$

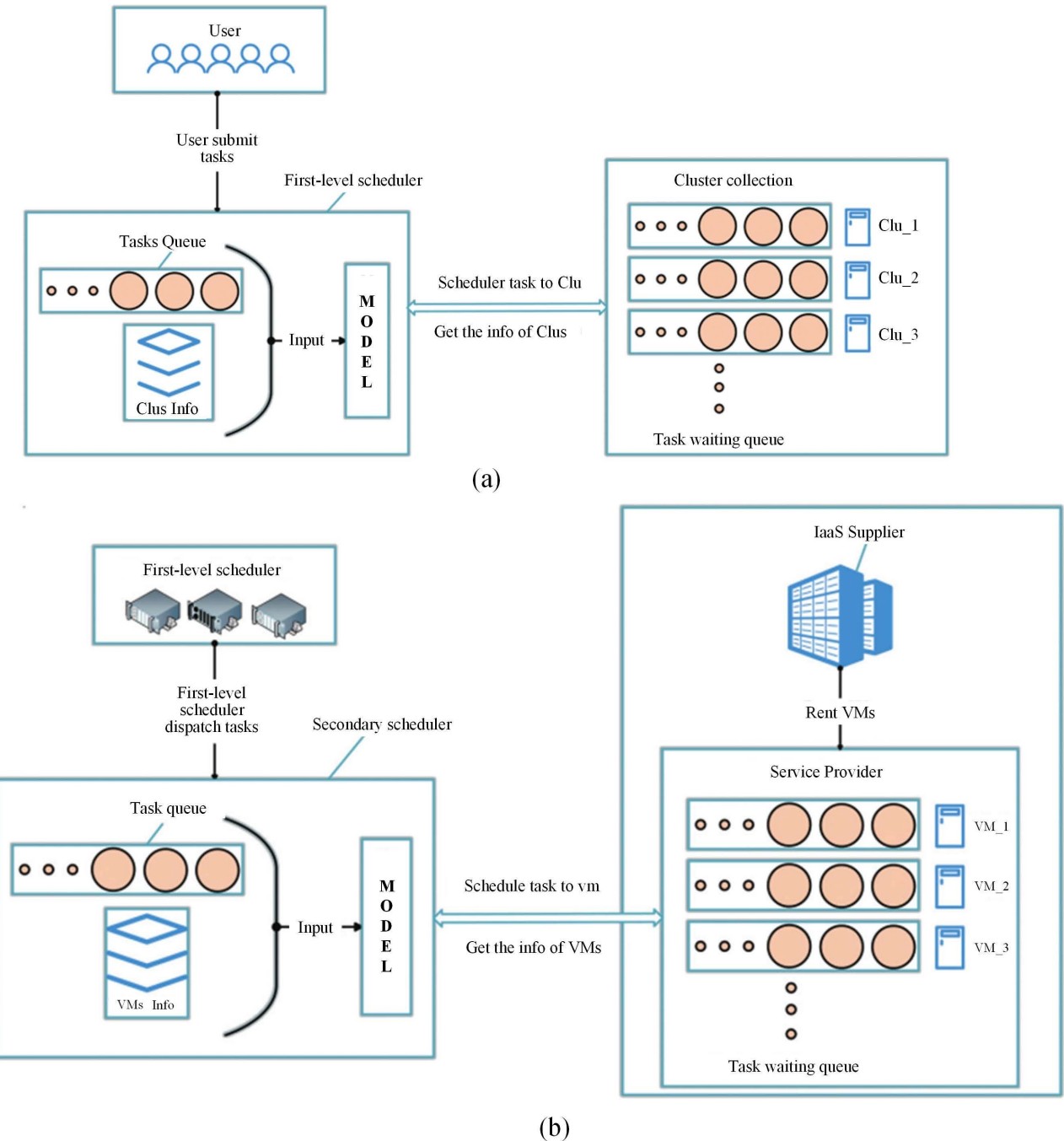

**Fig 2. Scheduling model.** (a) first layer; (b) second layer.

where i is the task number; $t_i^{mips}$ and $t_i^{bw}$ represent the task's requirements for instruction processing speed and bandwidth resources respectively; $t_i^{mips-l}$ and $t_i^{bw-l}$ represent the number of instructions to be processed and the size of the transmitted data of the task respectively; $t_i^{dead} = \frac{t_i^{mips-l}}{t_i^{mips}} + \frac{t_i^{bw-l}}{t_i^{bw}}$ is the task processing time expected by the user; and $t_i^{load}$, $t_i^{start}$, $t_i^{finish}$, and $t_i^{cost}$ specify the time when the task is submitted to the dispatch center, the time when the task is executed by the

node, the time when task execution is complete, and the corresponding cost, respectively. Owing to differences in processing capabilities among VMs, the costs are diverse, so the final cost of a task is determined according to the VM and the execution time.

In the first-level scheduler, a cluster is a collection of several VMs. A cluster and the VMs contained in it are represented by clu and vm, respectively, and given by the following:

$$vm_k = \{vm_k^{mips}, vm_k^{bw}, vm_k^{price}, vm_k^{busytime}, vm_k^T\} \tag{6}$$

where $k$ is the number of the VM, $vm_k^{mips}$ is the number of instructions executed by $vm_k$ per second, $vm_k$ is the bandwidth of $vm_k$, $vm^T$ is the task buffer queue corresponding to $vm_k$, and $vm_k^T$ temporarily stores its assigned tasks from the first-level scheduler. We use $\left|vm_k^T\right|$ to denote the number of tasks in the buffer queue, and $vm_k$ can execute tasks only if

$$t_i^x \leq vm_k^x \ \forall x \in \{mips, bw\} \tag{7}$$

where $vm_k^{price}$ is the price of $vm_k$ per second, which we set according to Alibaba Cloud resource pricing. For computing resources, the pricing is linear, and bandwidth resources are set to tiered pricing. We use price ratios (0.0003/mips/s, (0.063, 0.248)/mb/s), and calculate the prices of vm as

$$
\begin{aligned}
vm_{mips}^{price} &= vm^{mips} * mips_{price} \\
vm_{bw}^{price} &= \begin{cases} vm^{bw} * bw_{price1} & (vm^{bw} < bw_{basic}) \\ bw_{basic} * bw_{price1} + (vm^{bw} \\ \quad -bw_{basic}) * bw_{price2} & (else) \end{cases} \\
vm^{price} &= vm_{mips}^{price} + vm_{bw}^{price}
\end{aligned}
\tag{8}
$$

where $bw_{basic}$ is the pricing demarcation point of the bandwidth step price; $bw_{price(0.063)}$ and $bw_{price2(0.248)}$ are the first- and second-stage bandwidth prices, respectively; the price exceeding $bw_{basic}$ is calculated using $bw_{price}$; and $mips_{price}$ is the price of computing resources. $vm_{bw}^{price}$ are different calculation formulas depending on whether they are greater than a certain step value.

We define the execution time of $vm_k$ to execute task $t_i$ as

$$ET_{ik} = \frac{t_i^{mips-l}}{vm_k^{mips}} + \frac{t_i^{bw-l}}{vm_k^b w}(t_i^{mips} \leq vm_k^{mips}, t_i^{bw} \leq vm_k^{bw}) \tag{9}$$

that is, the sum of the ratios of workloads of resources that task $vm_k$ needs to handle and the processing capabilities of $vm_k$. Here, the SLA requirements need to be met ($t_i^{mips} \leq vm_k^{mips}, t_i^{bw} \leq vm_k^{bw}$).

We define $vm_k$ to complete all assigned tasks under the current time clock as:

$$vm_k^{busytime} = \sum_{i=1}^{|vm_k^T|} ET_i k - clock + t_1^{start} \tag{10}$$

where $clock$ is the current moment, that is, the time when the decision is made, and where $vm_k^T$ is the task buffer queue at the current time clock, which stores all the tasks that $vm_k$ needs to complete. The head of the buffer queue is the task being processed by $m_k$, and its start execution time is set to $t_1^{start}$.

We define the task overdue time as the interval between the task completion time and the expected completion time after task $t_i$ is handed over to $vm_k$ for execution, that is,

$$RT_{ik} = vm_k^{busytime} + ET_{ik} + clock - t_i^{load} - t_i^{dead} \tag{11}$$

We determine the start time of task $t_i$ in $vm_k$ through $vm_k^{busytime}$ and the clock. The start time plus the execution time of task $ET_{ik}$ is the actual completion time of task $t_i$ in $vm_k$. The task submission time $t_i^{load}$ plus the task expected completion time $t_i^{dead}$ is the expected completion time.

After submission, a task is scheduled by the first-level scheduler to cluster $clu$, which is a collection of VMs, with related attributes

$$clu_j = \left\{ clu_j^{mips_{sum}}, clu_j^{bw_{sum}}, clu_j^T, clu_j^{busy_{VM}}, clu_j^{VM} \right\} \tag{12}$$

where $clu^{VM}$ consists of all VMs included in the cluster. The sums of the computing power and bandwidth of all VMs in clu are $clu_{sum}^{mips}$ and $clu_{sum}^{bw}$, respectively, which are calculated as

$$clu^{x_{sum}} = \sum_{k=1}^{|clu^{VM}|} vm_k^x (x \in mips, bw) \tag{13}$$

Similarly, we define the sum of the computing power and bandwidth of idle VMs in the cluster as

$$clu^{I-x_{sum}} = \sum_{k=1}^{|\complement_{clu^{VM}} clu^{busyVM}|} vm_k^x (x \in mips, bw) \tag{14}$$

where $clu^T$ is the task buffer queue corresponding to the cluster, which stores the tasks dispatched by the first-level scheduler. $clu^{busyVM}$ is a collection of working VMs in the cluster, and the number of busy VMs is $|clu^{busyVM}|$.

$\complement_{clu^{VM}} clu^{busyVM}$ is the complement set of busy VMs among all VMs in clu, that is, the set of idle VMs. A busy VM is defined as

$$clu_j^{busyVM} = \{vm_i | vm_i \in clu_j^{VM}, |vm_i^T| \geq 0\} \tag{15}$$

We also define $clu_i^{et}$ and $clu^{wet}$, which are the estimated execution times of the task and the task buffer, respectively, relative to all resources of the cluster; and $clu_i^{lt}$, the estimated execution time of the task relative to the idle resources of the cluster. Here, $clu_i^{et}$ reflects the comprehensive processing capacity of the cluster, and is the task workload divided by the total processing capacity of the cluster,

$$clu_{ij}^{et} = \frac{t_i^{mips-I}}{clu_j^{mips_{sum}}} + \frac{t_i^{bw-I}}{clu_j^{bw_{sum}}} \tag{16}$$

where $clu^{wet}$ is the total number of tasks assigned to the cluster divided by the cluster capacity,

$$clu_j^{wet} = \sum_{i=1}^{|clu^T|} clu_{ij}^{et} \tag{17}$$

 

which can reflect the current load situation of the cluster to a certain extent.

$clu_i^{lt}$ measures the current cluster load and idle computing power in conjunction with the number of idle VMs, if any. It is the amount of the current task divided by the total processing capacity of the cluster's idle VMs,

$$clu_{ij}^{lt} = \frac{t_i^{mips-l}}{clu_j^{l-mips_{sum}}} + \frac{t_i^{bw-l}}{clu_j^{l-bw_{sum}}}$$

(18)

The average price of all VMs in the statistical cluster is

$$clu^{pricemean} = \frac{\sum_{k=1}^{|clu^{VM}|} vm_k^{price}}{|clu^{VM}|}$$

(19)

### 3.4. Algorithm design

Hierarchical DRL plays a crucial role in task scheduling in the cloud model. In the cloud environment, the scale of tasks and resources is vast and highly complex and variable. The hierarchical architecture divides task scheduling into two levels: the cluster level and the virtual machine level. At the cluster level, through the DRL algorithm, the suitability of tasks and clusters is evaluated and decisions are made. It can quickly screen out clusters with appropriate resource combinations (such as computing power, bandwidth, etc.), avoiding a global search among all VMs, thereby significantly reducing the complexity of the decision space and improving the initial efficiency of task allocation. At the virtual machine level, DRL further makes refined scheduling decisions for the VMs within the cluster. Taking into account dynamic information such as the real-time load and task processing history of the VMs, it determines the specific virtual machine that is most suitable for executing the task. The combination of this hierarchical structure and DRL enables the scheduler to gradually optimize the scheduling strategy in a large-scale cloud environment according to the dynamic changes of tasks and resources, achieving efficient task allocation and resource utilization and effectively addressing the challenges of complexity and dynamism in task scheduling under the cloud model. We introduce a hierarchical intelligent decision-making approach using Double Deep Q-Networks (DDQN) to tackle the challenge of large-scale online task scheduling in dynamic cloud computing environments. The design of the state and action spaces is crucial for the effectiveness of our model in adapting to this dynamic environment, as illustrated in Fig 3. Specifically, we represent the state space using a 12-dimensional vector, which encapsulates information about task-cluster combinations and environmental characteristics. This vector is processed to ensure fixed dimensions through the use of average and standard deviation calculations, thereby accommodating the variability inherent in the environment. Additionally, we define the action space by combining VMs and clusters, allowing the model to make informed decisions on task allocation. Each VM or cluster corresponds to a separate model, enabling parallel computation and facilitating efficient adaptation to changes in the environment. The output of each model indicates the degree of compatibility between the task and the corresponding VM or cluster, facilitating the selection of the most appropriate combination for task execution.

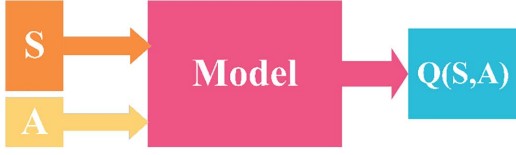

**Fig 3. Q value model.**

**3.4.1. First-level scheduler input vector.** We represent the task-cluster combination and environment information as a 12-dimensional vector. Through the use of average and standard deviation processing for dynamic environmental characteristics, a vector with uncertain dimensions is transformed into a vector with fixed dimensions.

We define the model input set for the first-level scheduling decision of task $t_i$ as $CS_i = \{cs_{ij}|clu_j \in CLU\}$, where the CLU is the set of all clusters, and where $cs_{ij}$ is an 18-dimensional vector,

$$cs_{ij} =$$
$$\left\{
\begin{array}{c}
|clu_j^{busyVM}|, Mean(\{|clu_{j2}^{busyVM}|\}), \\
Std(\{|clu_{j2}^{busyVM}|\}), clu_j^{pricemean}, \\
Mean(\{clu_{j2}^{pricemean}\}), Std(\{clu_{j2}^{pricemean}\}), \\
clu_{ij}^{et}, Mean(\{clu_{ij2}^{et}\}), Std(\{clu_{ij2}^{et}\}), \\
clu_{ij}^{price}, Mean(\{clu_{ij2}^{price}\}), Std(\{clu_{ij2}^{price}\}), \\
clu_j^{wet}, Mean(\{clu_{j2}^{wet}\}), Std(\{clu_{j2}^{wet}\}), \\
clu_{ij}^{lt}, Mean(\{clu_{ij2}^{lt}\}), Std(\{clu_{ij2}^{lt}\}), \\
|clu_{j2} \in CLU
\end{array}
\right\}$$

(20)

where $Mean(X)$, $Std(X)$, and $Min(X)$ are the average, standard deviation, and minimum value, respectively, of the set X. We define the input of the decision of each task $t_i$ as a CS set. There are several 18-dimensional CS sets, each corresponding to a decision model. In the formula, $\left|clu_j^{busyVM}\right|$ is the number of idle VMs in the cluster, $clu_j^{et}$ is the total processing time, $clu_j^{wet}$ is the processing time of tasks to be allocated in the cluster, and $clu_j^{lt}$ is the processing time of idle VMs, which is zero when there is no idle VM. The average rate of task $t_i$ in cluster $clu_j$ is $clu_{ij}^{price} = clu_j^{pricemean} * clu_{ij}^{et} * \left|clu_j^{busyVM}\right|$.

**3.4.2. Secondary scheduler input vector.** We represent the combination of task–VM and environment information as a 20-dimensional vector. The previous average value and standard deviation are used to process the environmental features, and the minimum value is added to convert the original dynamic vector dimension to a fixed vector dimension.

We define the model input set for the second-level scheduling decision of task $t_i$ as $VS_i = \{vs\_ik|vm_k \in clu^{VM}, t_i^{mips} \le vm_k^{mips}, t_i^{bw} \le vm_k^{bw}\}$, where $clu^{VM}$ is the VM set of the scheduled cluster, and where $vs_{ik}$ is a 20-dimensional vector,

$$\left\{
\begin{array}{c}
vm_k^{busytime}, Mean(\{vm_{k2}^{busytime}\}), \\
Std(\{vm_{k2}^{busytime}\}), Min(\{vm_{k2}^{busytime}\}) \\
ET_{ik}, Mean(\{ET_{ik2}\}), Std(\{ET_{ik2}\}), Min(\{ET_{ik2}\}) \\
ET_{ik} * vm_k^{price} Mean(\{ET_{ik2} * vm_k 2^{price}\}), \\
Std(\{ET_{ik2} * vm_k 2^{price}\}), Min(\{ET_{ik2} * vm_k 2^{price}\}) \\
max(RT_{ik}, 0), Mean(\{max(RT_{ik2}, 0)\}), \\
Std(\{max(RT_{ik2}, 0)\}), Min(\{ET_{ik2} * vm_k 2^{price}\}) \\
RT_{ik}, Mean(\{RT_{ik2}\}), Std(\{RT_{ik2}\}), Min(\{RT_{ik2}\}) \\
|vm_{k2} \in VM
\end{array}
\right\}$$

(21)

Five basic elements are included in $vs_{ik}$. $vm_k^{busytime}$ is the estimated busy time of $vm_k$ (Equation (7)). $ET_{ik}$ is the time it takes for task $t_i$ to execute in $vm_k$. $ET\_ik * vm_k^{price}$ is a cost-related feature. $RT_{ik}$ is the estimated overdue time for task $t_i$ to execute on $vm_k$ (Equation (8)). When there is no overdue time, $RT_{ik}$ is less than or equal to 0, which indicates the degree of surplus resources in that case, and $max(RT_{ik}, 0)$ represents the degree of overdue time when the task is overdue. We consider the average value, standard deviation, and minimum value of these five characteristic values of the same task $t_i$ under different VMs as environmental information.

**3.4.3. Reward function.** Our scheduling environment is subject to the condition that the load level can be changed, so we prioritize the task overdue time as the primary goal to ensure the user experience, and we optimize the task cost on this basis.

However, these goals contradict each other. For example, choosing a low-cost priority strategy will inevitably increase the utilization rate of low-configuration VMs, which will cause more time to be spent. To reduce the task overdue time as the main optimization goal, the utilization rate of the high-profile VM will correspondingly increase, thereby increasing the fee rate. Therefore, for multiobjective optimization problems, it is necessary to balance the importance of objectives, and the dimensions of goals are best if they are consistent. However, owing to the dynamic nature of the environment, adjusting the dimensions to achieve complete consistency is difficult. Therefore, we adjust only the relevant parameters to keep the target magnitudes close.

Our work in the reward function has two goals. Under a high load, we focus on the user experience and attempt to reduce the task overdue time. Under medium and low loads, there is a tradeoff between cost and task overdue time. Therefore, we define the reward function as

$$r_i = r1_{rate} * max(1 - r2, 0.1) - r2 \tag{22}$$

The calculations of r1 and r2 are related to the task cost and task overdue time, respectively. In a high-load environment, $r2$ is larger, that is, the single task overdue time index is greater than 1. Hence, the main optimization goal of scheduling is to reduce r2, that is, to reduce the task overdue time, so the proportion of r1 in the return is reduced to an extremely low level. When the load reduces to low or medium, the task overdue time is between 0 and 1, the return is composed of r1 and r2, and their proportions are dynamically adjusted according to the load pressure. When r2 is 0, resources are in surplus, and the return is composed entirely of $r1$; that is, the main optimization target is based on the rate.

r1 is the task rate of this batch of tasks, which is calculated by dividing the total of the task calculation and transmission volume of this batch by the total cost of this batch of tasks,

$$r1_i = \frac{\sum_{k=1}^{|T|} (t_k^{mips-l} + t_k^{bw-l})}{|T| * \sum_{k=1}^{|T|} t_k^{expend}} * \alpha_{r1} \tag{23}$$

where T is the task set of the task buffer queue of the current batch, $|T|$ is the number of tasks in this batch, the numerator is the sum of the calculation and transmission of all tasks in this batch, and $a_{r1}$ is the adjustment parameter for the overall scaling of r1. Since the cost and the number of tasks are positively correlated, we use the proportional band between the task amount and cost to express the return of the cost, rather than simply using the cost value.

r2 is the task overdue time, that is, the task completion time exceeding the user's expected completion time,

$$r2_i = max(t_k^{finish} - t_k^{load} - t_k^{dead}, 0) * \alpha_{r2} \tag{24}$$

The reward of the task overdue time is calculated by the $max$ function, which intercepts the part with an overdue time greater than 0 as a penalty and does not treat the early completion time as a reward. This will guide the model to schedule tasks to more suitable VMs instead of greedily scheduling tasks to the best performing VMs. $a_{r2}$ is an adjustment parameter for the overall scaling of r2.

In this study, $a_{r1} = \frac{1}{20000}$, $a_{r2} = \frac{1}{100}$. These parameters are used to make the magnitudes of r2 and r1 similar, and to scale the overall return.

## 4. Results and discussion

### 4.1. Evaluation platform and benchmarks

To facilitate training, we built a simulation environment in the Python language to imitate CloudSim [33], which is the most commonly used cloud computing simulation open-source toolkit. Task scheduling was assigned to six clusters, with a total of 162 VMs. The model training parameter settings are shown in Table 1.

**Table 1. DQN training parame.**

| Parameter | First-level | Second-level |
|---|---|---|
| Gamma | 0.9 | 0.9 |
| Learning rate | data | data 1 |
| Loss function | smooth L1 | smooth L1 |
| Mini-batch | 50 | 50 |
| Optimizer | Adam | Adam |
| Replay memory | 50000 | 10000 |

To test the performance of our proposed task scheduling framework, the experiment was tested on Google workload tracking and randomly generated workloads as benchmarks. A randomly generated benchmark automatically generates the workload type according to the set parameters, and generates an arrival time according to the Poisson distribution and set arrival rate. The parameters include the number of tasks in each batch, the duration of a task, the bottom line, and the floating range of the task's resource requirements. The settings are shown in Table 2.

Google's workload tracking is a segment of the usage trajectory of the Google cluster. However, the record does not contain the task length, so this was calculated by the continuous execution time of the task, the average CPU utilization rate, and the CPU processing capacity as

$$I_g = (tg_{finish} - tg_{start}) * U_{avg} * C_{CPU}$$

(25)

where $tg_{start}$ and $tg_{finish}$ are the timestamps of the start and end of task tracking, respectively; $U_{avg}$ is the average CPU utilization of the task; and $C_{CPU}$ is the processing power of the CPU. Since this variable is not given in Google cluster usage tracking, we assume that it is similar to that of the host used in our experiments, $C_{CPU}$= 1 million instructions per second (MIPS) [34].

## 4.2. Comparison algorithms and the evaluation index

The algorithms for comparison used the following first-level scheduling strategies: random (scheduling jobs to random clusters), round- robin (sequential allocation, assigning jobs to clusters in a polling manner), Min-Min (always finding the task with the smallest load, and dispatching it to the earliest completed computing node), and dominant resource fairness (DRF) (a general multiresource allocation strategy). The Min-Min strategy is uniformly used in second-level scheduling, with better comprehensive performance than the other comparison algorithms.

We used the following performance indicators in the evaluation.

The makespan is the completion time of the last task. The task cost is the sum of the execution times of tasks in the corresponding VM multiplied by the VM rate.

Load balancing (LB) adds the load balancing value of each batch of tasks as the load balancing index of the method, and the load balancing value is the sum of the standard deviations of the resource utilizations of each cluster.

**Table 2. DQN training parame.**

| Parameter | Rang |
|---|---|
| num | num [2,36) |
| mips | mips [100,4000) |
| bw | bw [40,250) |
| duration | [5,30) |

Overdue time is the difference between the completion time and loading time of each task. If the overdue time is less than 0, we truncate it to 0 as the ReLU function.

### 4.3. Experiment on a randomly generated benchmark

We used a randomly generated benchmark to compare the performance of the four algorithms under different loads. By adjusting the task arrival rate to control the load level, the task arrival rate was set to 0.1, 0.4, 0.8, and 3.0. We used 50 experimental results for analysis, and drew a comparison chart of the average value and a 4-point map, in which the red line corresponds to the average value.

Overdue time: Fig 4a shows the experimental results for the task overdue time indicator. Under a low load, HITS prefers a strategy with a low cost that can guarantee a lower task overdue time. As shown in Fig 4a, under a low load, the overdue time of the comparison algorithms is 0, but HITS still results in a small amount of time. According to the design of the reward function, an overdue time less than 1 is considered acceptable, and the cost goal makes the scheduling strategy more inclined to assign tasks to a VM with a lower cost. Under medium and high loads, the HITS will seek to maximize utilization, which will reduce its task overdue time. In general, resources are dynamically requested and released to reduce the number of overload situations. However, Fig 4b shows the volatility of the HITS. As the load increases, its fluctuation range begins to expand, and a small number of abnormal points appear, but its average performance is better than that of the comparison algorithms.

Cost: Fig 5a shows the experimental results of the cost index. Under a low load, HITS prefers lower-cost machines. As the load increases, HITS trades off between cost and processing speed, and will be punished when a task is overdue. This makes HITS choose a low-cost machine that can achieve the required processing speed. Under a high load, HITS takes task overdue time as the main optimization goal, and the strategy will allocate tasks to suitable machines to improve machine utilization and throughput, reduce task completion time, and reduce costs. In the comparison algorithms, DRF and Min-Min select the machine that can complete the task the fastest, so these algorithms cannot fully utilize all VMs. The random and round robin approaches are distributed more evenly, which increases some low utilization rates. However this also leads to a longer execution time, thus leading to a weak advantage in cost. Fig 5b shows that HITS has a certain degree of volatility, which increases with the optimization space, but it still has advantages over the comparison algorithms.

LB: Fig 6a shows the experimental results for the load balancing index. Under a low load, the low-cost strategy adopted by HITS is inclined to allocate tasks to low-cost machines, making the distribution unbalanced. However, as the load increases, the proportion of task overdue time in the main optimization goal of HITS begins to increase. The HITS begins to choose a strategy with a higher utilization rate, which shows a clear advantage on the LB. Owing to the large differences in the processing capabilities of clusters, the random and round robin approaches do not consider the cluster processing capabilities, but choose feasible clusters, so they perform poorly on the LB index, whereas DRF and Min-Min are allocated according to the processing capabilities of the cluster, so the overall LB performance is better. HITS performs better under medium and high loads. As shown in Fig 6b, HITS fluctuates greatly under low loads, because its strategy does not consider equilibrium at this time. However it is mainly based on cost. As the load increases, the HITS adopts a high utilization strategy, which indirectly leads to a decrease in the LB index.

Makespan: Fig 7a shows the experimental results under the makespan indicator. Under a low load, HITS chooses more machines with low processing power, so task overdue time is generated, and task completion times are greater. Since the makespan indicator is the completion time of the last task, there is idle time between each batch of tasks under a low load, so the difference between the comparison algorithms comes from the decision result of the last batch of tasks; hence it is smaller and does not show obvious advantages. As the load increases, Min-Min and DRF assign tasks on basis of the the capabilities of each cluster to achieve higher utilization, and HITS chooses a higher utilization allocation strategy so that tasks can be completed as soon as possible, with obvious advantages in makespan. Fig 7b shows the volatility of the HITS. Although it is not obvious, there are still abnormal points.

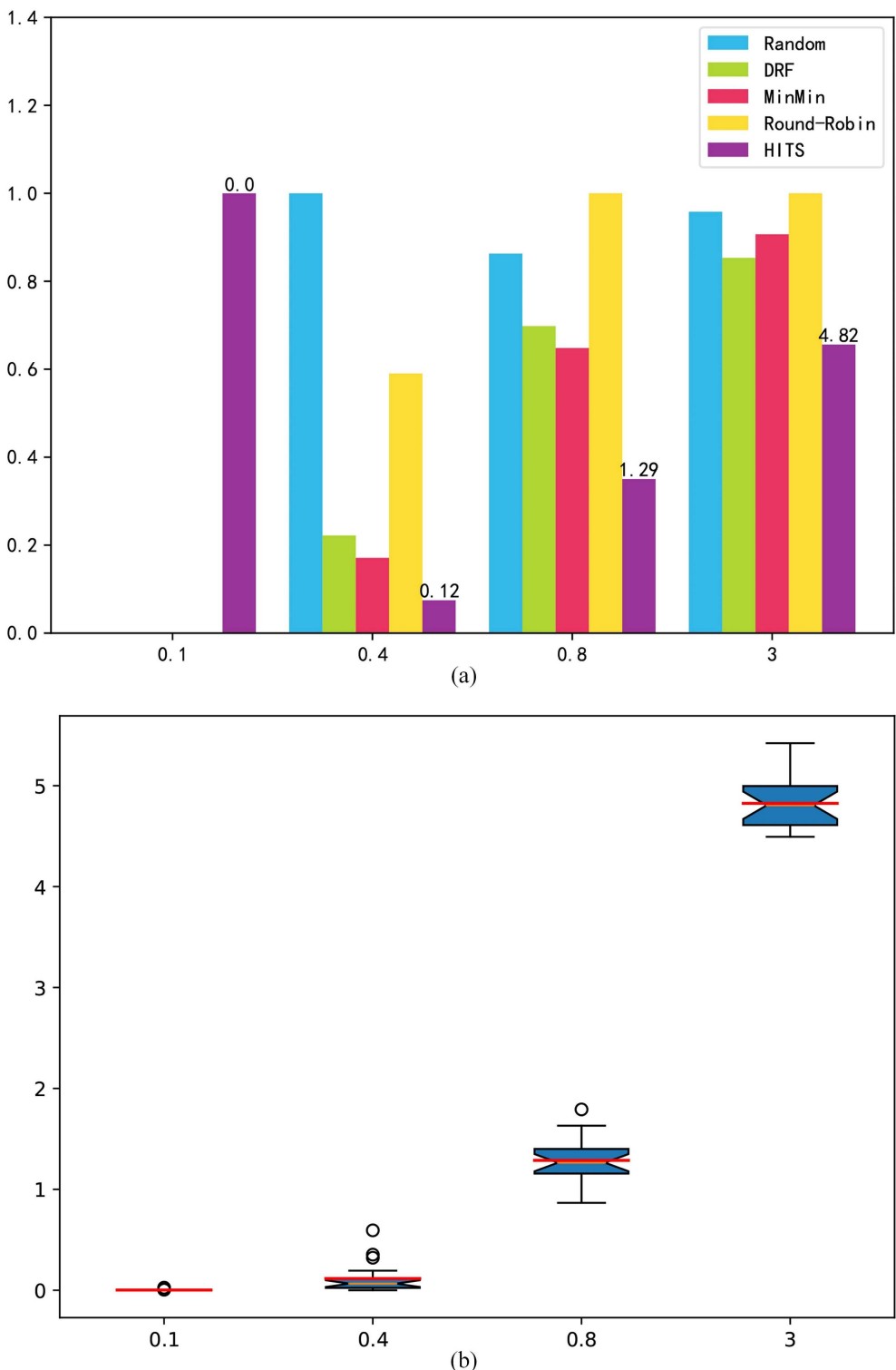

**Fig 4. Randomly generated benchmark overdue time.** (a) mean result; (b) box-plot.

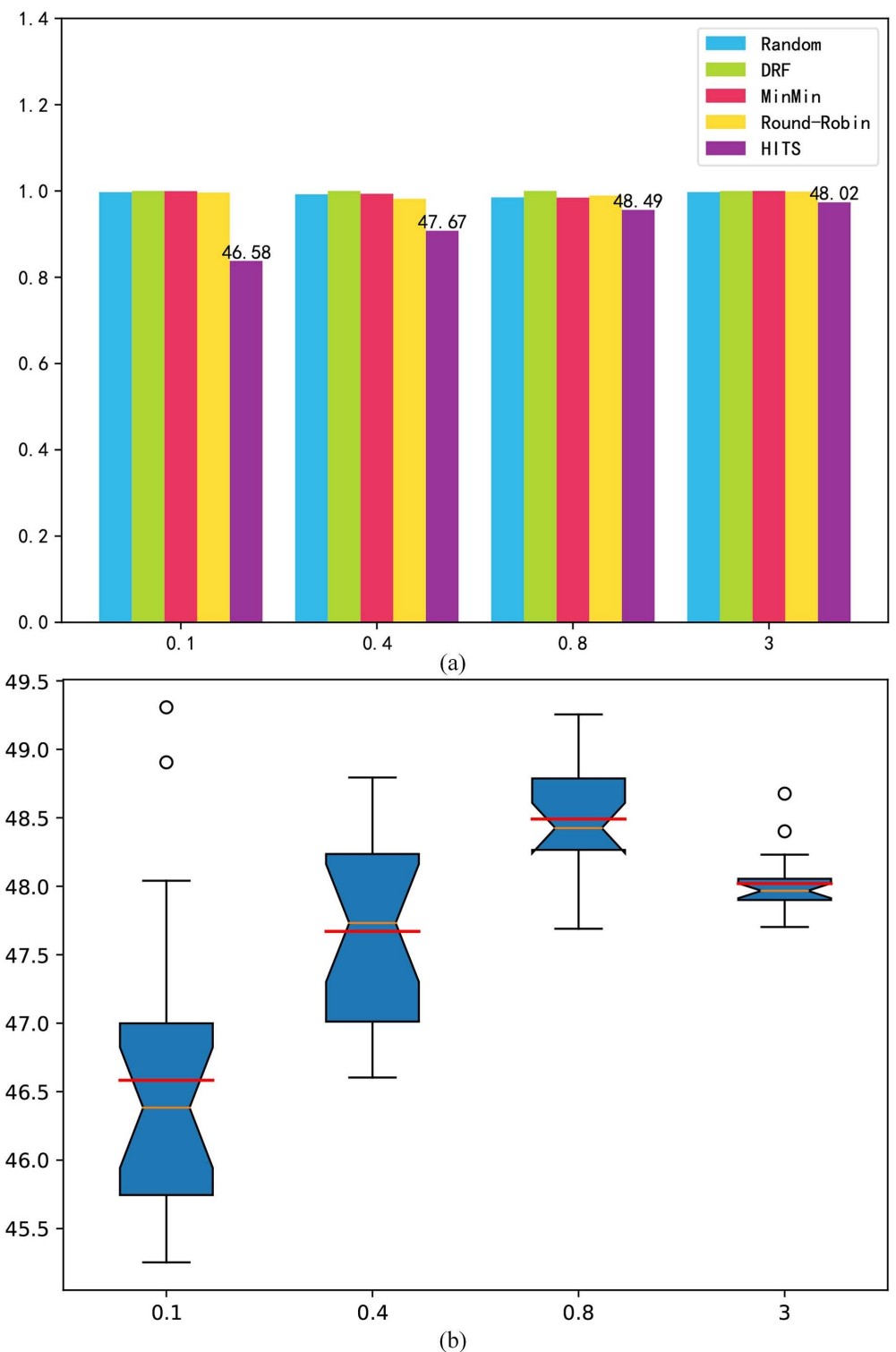

**Fig 5. Randomly generated benchmark cost index.** (a) mean result; (b) box-plot.

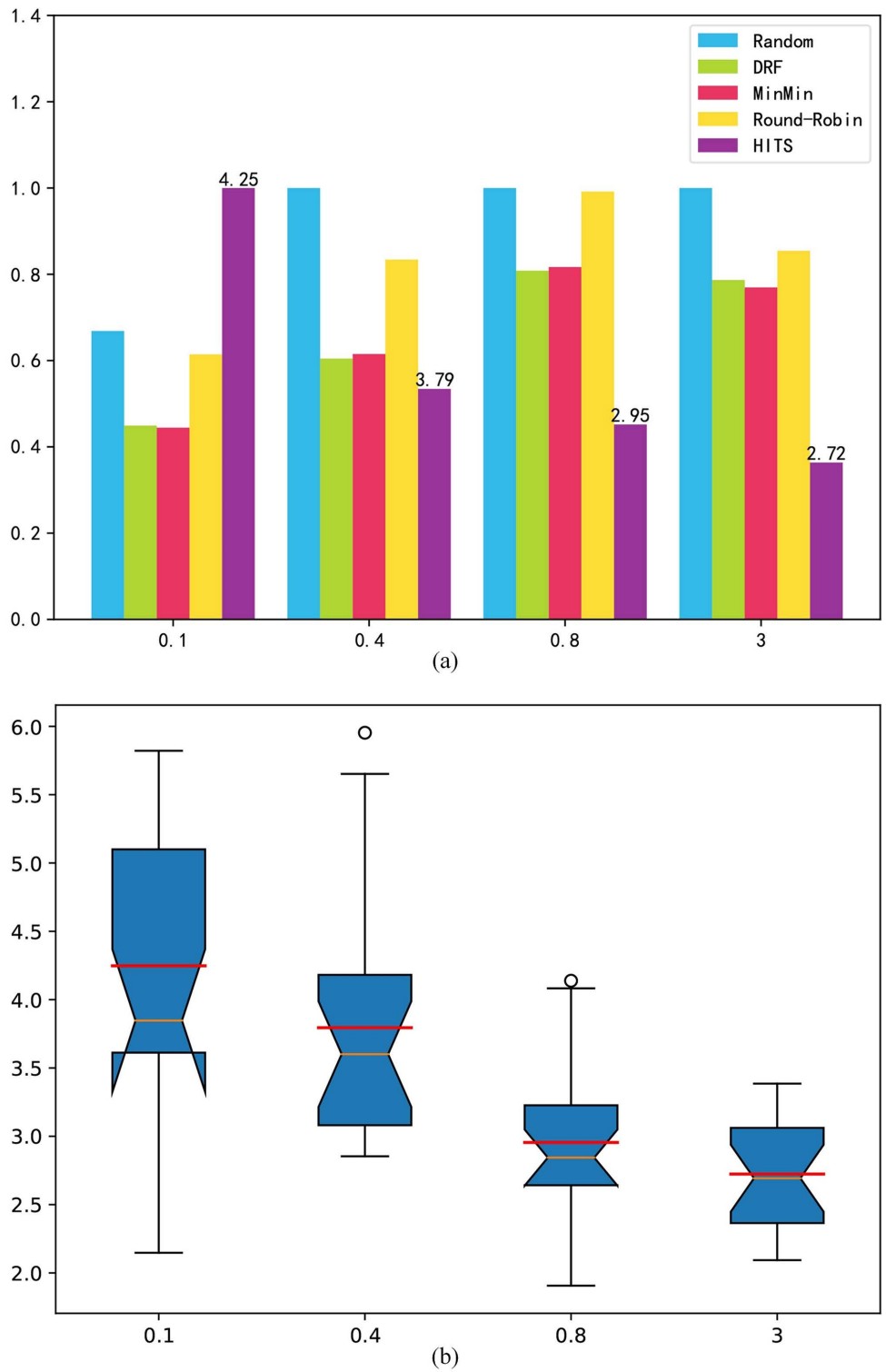

**Fig 6. Randomly generated benchmark load balancing.** (a) mean result; (b) box-plot.

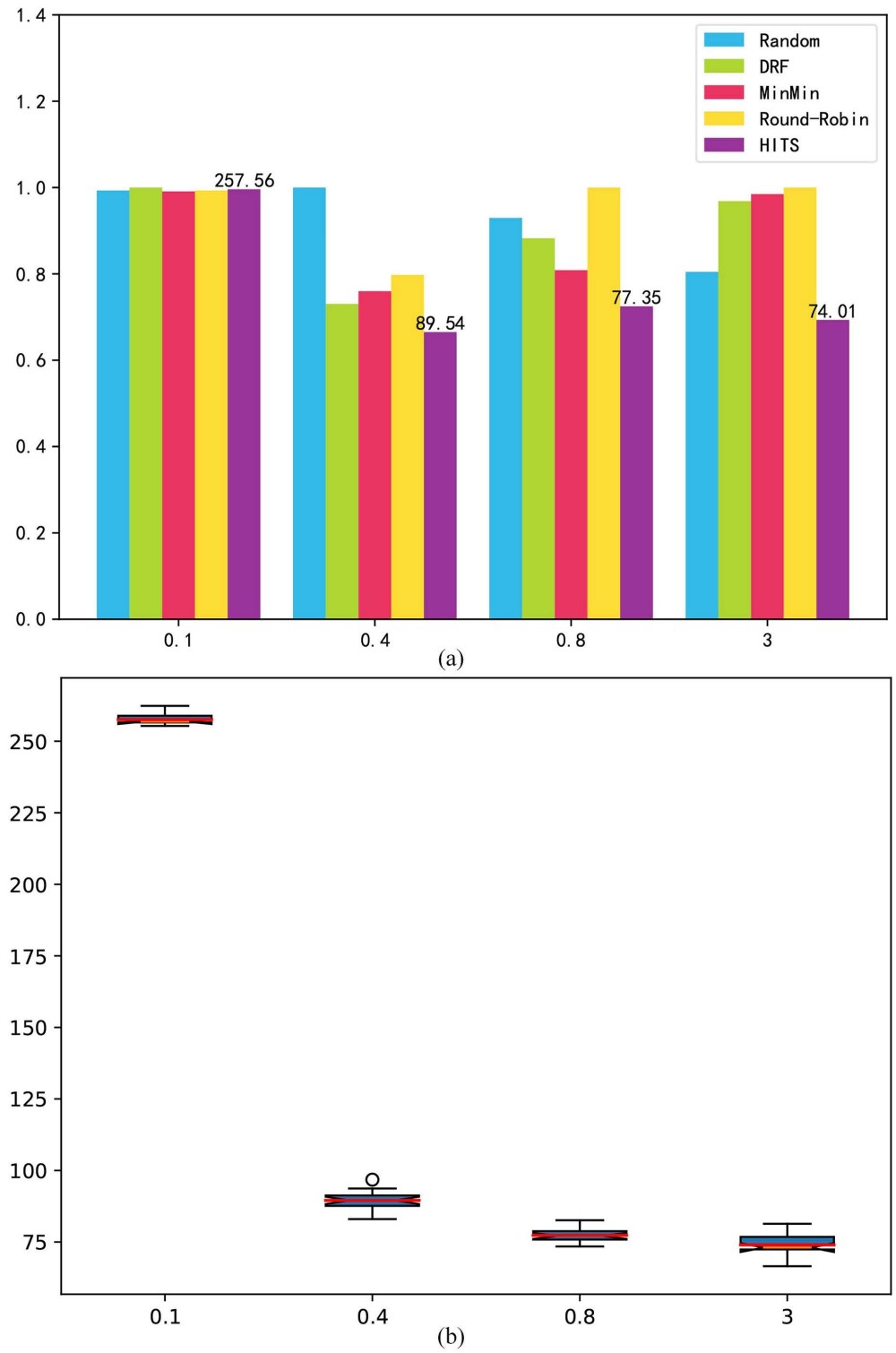

**Fig 7. Randomly generated benchmark makespan.** (a) mean result; (b) box-plot.

### 4.4. Experiments on the Google workload benchmark

Fig 8 shows the performance of the algorithm on the Google workload benchmark. The HITS has high volatility and many abnormal points, that is, the degree of fluctuation and optimization of the algorithm are affected by the environment (intensity and type of load). HITS can still achieve excellent performance under different work types and loads, but does so in relation to the abnormal frequency of decision-making.

In summary, HITS can switch between strategies according to the load, and shows better comprehensive performance than the comparison algorithms do on each index. However, owing to the characteristics of the algorithm, the effect is not stable enough, and there are certain abnormal scheduling situations. The situation is within the acceptable range, but there are a few unacceptable abnormalities. In actual situations, certain heuristic strategies can be considered to reject decisions with large abnormalities in advance.

## 5. Conclusions

With the increasing popularity of cloud computing services, their large and dynamic load characteristics have rendered task scheduling an NP-complete problem. We propose a hierarchical framework for large-scale online task scheduling to reduce task cost and overdue time, which utilizes the hierarchical DRL approach. It groups VMs into clusters and effectively decomposes the complex task scheduling problem through hierarchical partitioning. A scheduler based on deep reinforcement learning is employed, and its return function flexibly trades off and optimizes the task overdue time and cost according to the load dynamics of the cloud environment. This enables the scheduler to automatically learn and adopt the most appropriate scheduling strategy under different load conditions, significantly enhancing the intelligence level of task scheduling.

The uniqueness of this framework lies in the meticulously constructed state space and return function. The state space effectively models the uncertainties of the cloud environment by integrating multidimensional information of tasks, clusters, and VMs (such as resource requirements, processing capabilities, and cost), and applying statistical methods such as averaging and standard deviation to handle the characteristics of the dynamic environment. The return function flexibly trades off and optimizes between task overdue time and cost according to the load dynamics of the cloud environment, enabling the scheduler to automatically learn and adopt the most appropriate scheduling strategy under different load conditions, significantly enhancing the intelligence level of task scheduling.

Experiments demonstrate that it skillfully balances cost and performance. In low-load situations, costs are reduced by using low-cost nodes within the service level agreement (SLA) range; in high-load situations, resource utilization is improved through load balancing. Compared with classical heuristic algorithms, it effectively optimizes load balancing, cost, and overdue time, achieving a 10% overall improvement. Compared with a variety of traditional task scheduling algorithms (such as random, round robin, Min-Min, and dominant resource fairness), the HITS framework has distinct advantages in terms of multiple key performance indicators. In terms of load balancing, it can effectively reduce the difference in resource utilization among clusters; in cost control, it achieves a significant reduction in task execution cost; in terms of task execution time, it greatly shortens the average completion time of tasks, thus providing a more efficient, intelligent, and reliable solution for cloud computing task scheduling.

There are still shortcomings in the method used in this study. First, the continuous learning and updating of network parameters might introduce latency, which could impact real-time task scheduling efficiency. Furthermore, the framework's performance heavily depends on the quality and quantity of training data, which might be challenging to obtain and maintain in a dynamic cloud environment.

Our future work will focus on improving the decision-making stability of the algorithm. This approach will combine user requests and the dynamic supply of resources with the DRL method to solve the problem of task scheduling and dynamic resource adjustment to meet other goals.

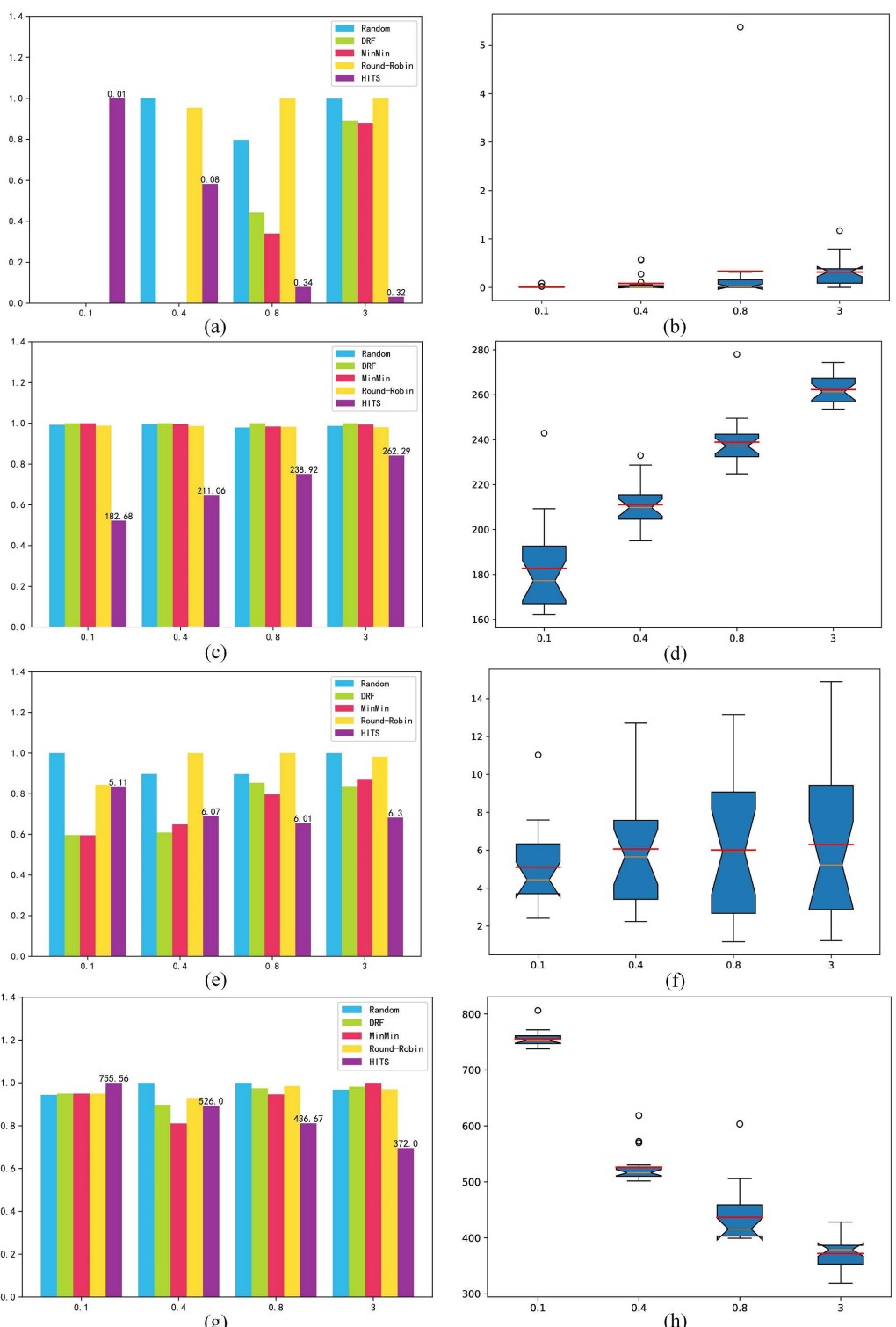

**Fig 8. Google benchmark results.** (a) overdue time mean result; (b) overdue time box-plot; (c) cost index mean result; (d) cost index box-plot; (e) load balancing mean result; (f) load balancing box-plot; (g) makespan mean result; (h) makespan box-plot.

## Author contributions

**Conceptualization:** Delong Cui, Zhiping Peng.

**Formal analysis:** Qirui Li, Jieguang He.

**Funding acquisition:** Delong Cui, Zhiping Peng, Qirui Li, Xiangwu Deng.

**Investigation:** Delong Cui.

**Methodology:** Zhiping Peng, Kaibin Li.

**Resources:** Delong Cui.

**Supervision:** Zhiping Peng.

**Validation:** Delong Cui, Kaibin Li, Qirui Li.

**Visualization:** Jieguang He.

**Writing – original draft:** Delong Cui.

**Writing – review & editing:** Zhiping Peng, Xiangwu Deng.

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
