## [Decision Letter · Decision Letter 0]

19 Nov 2024

PONE-D-24-45416An novel cloud task scheduling framework using hierarchical deep reinforcement learning for cloud computingPLOS ONE

Dear Dr. cui,

Thank you for submitting your manuscript to PLOS ONE. After careful consideration, we feel that it has merit but does not fully meet PLOS ONE’s publication criteria as it currently stands. Therefore, we invite you to submit a revised version of the manuscript that addresses the points raised during the review process.

We look forward to receiving your revised manuscript.

Kind regards,

Sameena Naaz

Academic Editor

PLOS ONE

2. Please note that PLOS ONE has specific guidelines on code sharing for submissions in which author-generated code underpins the findings in the manuscript. In these cases, all author-generated code must be made available without restrictions upon publication of the work. Please review our guidelines at https://journals.plos.org/plosone/s/materials-and-software-sharing#loc-sharing-code and ensure that your code is shared in a way that follows best practice and 

4. Thank you for stating the following financial disclosure: Key Realm R&D Pro-gram of Guangdong Province(2021B0707010003); National Natural Science Foundation of China (62273109); Guangdong Basic and Applied Basic Research Foundation (2022A1515012022, 2023A1515240020, 2023A1515011913); Key Field Special Project of Department of Education of Guangdong Province (2024ZDZX1034); Maoming Science and Technology Project (210429094551175, 2022DZXHT028, mmkj2020033); Projects of PhDs’ Start-up Research of GDUPT (2023bsqd1002, 2023bsqd1013, XJ2022000301); Special Innovation Projects for Ordinary Universities in Guangdong Province in 2023 (2023KTSCX086).  

Reviewers' comments:

Reviewer's Responses to Questions

**Comments to the Author**

1. Is the manuscript technically sound, and do the data support the conclusions?

Reviewer #1: Yes

Reviewer #2: Partly

2. Has the statistical analysis been performed appropriately and rigorously? 

Reviewer #1: Yes

Reviewer #2: No

3. Have the authors made all data underlying the findings in their manuscript fully available?

Reviewer #1: Yes

Reviewer #2: No

4. Is the manuscript presented in an intelligible fashion and written in standard English?

Reviewer #1: Yes

Reviewer #2: No

5. Review Comments to the Author

Reviewer #1: The paper proposes a novel cloud task scheduling framework using hierarchical deep reinforcement learning (DRL) to address the challenges of large-scale task scheduling in cloud computing. The framework defines a set of virtual machines (VMs) as a VM cluster and employs hierarchical scheduling to allocate tasks first to the cluster and then to individual VMs. The scheduler, designed using DRL, adapts to dynamic changes in the cloud environment by continuously learning and updating network parameters. Experimental results demonstrate that this approach effectively balances cost and performance, optimizing objectives such as load balance, cost, and overdue time. One potential shortcoming of the proposed hierarchical deep reinforcement learning (DRL) framework for cloud task scheduling could be its complexity and computational overhead. Implementing and maintaining a DRL-based scheduler requires significant computational resources and expertise in machine learning. Additionally, the continuous learning and updating of network parameters might introduce latency, which could impact real-time task scheduling efficiency.

How this could be handled in the proposed methodology?

Furthermore, the framework’s performance heavily depends on the quality and quantity of training data, which might be challenging to obtain and maintain in a dynamic cloud environment.

Contributions could be highlighted in the introduction sections

The quality, symmetry and the dimensions of figures needs to be improved and enhanced for formal presentation.

The methodology section is not distinguishing that how this particular technique is better than existing work.

The methodology section hardly cites any literature for the formulae and state-of-the-art techniques.

Some additional recommendations are as follows:

A. Paper needs to be revised for grammatical errors and typos.

B. Article needs to be proofread from the native English speaker or rewritten in the academic writing.

C. The abstract needs to showcase the numerical finding of the research study to reflect the contribution in the field.

D. The introduction needs to be providing rationale of the study, and brief literature review of existing studies, which is hard to differentiate in its current form.

Reviewer #2: 1 How the hierarchical deep reinforcement learning helps in scheduling of tasks in cloud paradigm?

2 Abstract should be concise and accurate.

3. What is the motivation behind cost, load balancing and how it will be handled by your scheduler and explain about its impact.

4. There are no contributions written by the authors in the article.

5. Motivation statements are not written in the manuscript.

6.Literature review is poor and research gap is not identified properly.

7.Deep Reinforcement learning technique was not properly mapped with respect to scheduling process.

8. Mathematical modelling is weak in the manuscript

9. Algorithm is design is not upto the mark

10.Result discussion is poor

6. PLOS authors have the option to publish the peer review history of their article (what does this mean? ). If published, this will include your full peer review and any attached files.

**Do you want your identity to be public for this peer review?** For information about this choice, including consent withdrawal, please see our Privacy Policy .

Reviewer #1: No

Reviewer #2: No

---

## [Author Response · Author response to Decision Letter 1]

19 May 2025

Reviewer #1:

Comment 1: The paper proposes a novel cloud task scheduling framework using hierarchical deep reinforcement learning (DRL) to address the challenges of large-scale task scheduling in cloud computing. The framework defines a set of virtual machines (VMs) as a VM cluster and employs hierarchical scheduling to allocate tasks first to the cluster and then to individual VMs. The scheduler, designed using DRL, adapts to dynamic changes in the cloud environment by continuously learning and updating network parameters. Experimental results demonstrate that this approach effectively balances cost and performance, optimizing objectives such as load balance, cost, and overdue time. One potential shortcoming of the proposed hierarchical deep reinforcement learning (DRL) framework for cloud task scheduling could be its complexity and computational overhead. Implementing and maintaining a DRL-based scheduler requires significant computational resources and expertise in machine learning. Additionally, the continuous learning and updating of network parameters might introduce latency, which could impact real-time task scheduling efficiency. How this could be handled in the proposed methodology? Furthermore, the framework’s performance heavily depends on the quality and quantity of training data, which might be challenging to obtain and maintain in a dynamic cloud environment.

Answer: Thanks for good advice. The shortcomings has revised in ABSTRACT and Conclusions.

ABSTRACT: There are still shortcomings in the method used in this article. Firstly, the continuous learning and updating of network parameters might introduce latency, which could impact real-time task scheduling efficiency. Furthermore, the framework's performance heavily depends on the quality and quantity of training data, which might be challenging to obtain and maintain in a dynamic cloud environment.

Comment 2: Contributions could be highlighted in the introduction sections

Answer: Thanks for good advice. Contributions has been highlighted in the introduction sections

This research proposes an innovative Hierarchical Intelligent Task Scheduling framework (HITS) based on the hierarchical deep reinforcement learning algorithm to address the challenge of large-scale task scheduling in cloud computing. Compared with traditional methods, HITS exhibits significant advantages. Firstly, through hierarchical partitioning and effective manipulation of the solution space, it accelerates the task scheduling process and simultaneously optimizes the task overdue time and cost, which is particularly crucial in large-scale task scheduling scenarios. Secondly, the model structure and return function of deep reinforcement learning are meticulously designed in accordance with the dynamic characteristics of the cloud environment. In response to the dynamic variation in the number of virtual machines, by modeling the Gaussian distribution of relevant features and using it as state information, the model can adaptively adjust. For different load conditions, a unique reward function is designed, which feeds back rewards based on the load to drive the model to learn corresponding decision-making strategies, thereby achieving efficient and intelligent task scheduling in the complex and variable cloud environment.

Comment 3: The quality, symmetry and the dimensions of figures needs to be improved and enhanced for formal presentation.

Answer: Thanks for good advice. The figures throughout the text have been revised.

Comment 4: The methodology section is not distinguishing that how this particular technique is better than existing work.

Answer: Thanks for good advice. The following content has been added to Chapter Three of Materials and methods.

Advantages of the Hierarchical Deep Reinforcement Learning Technology Employed in This Research over Existing Works:

Advantages of the Hierarchical Architecture: Compared with traditional single-layer task scheduling methods, the hierarchical architecture of HITS can effectively reduce the complexity of the problem. By dividing the task scheduling process into two levels, name-ly the cluster level and the virtual machine level, the decision space at each level is dimin-ished, and the scheduling efficiency is enhanced. In cluster-level scheduling, clusters suitable for task processing can be rapidly screened out, avoiding a global search among all virtual machines, thereby significantly shortening the task allocation time. Meanwhile, this hierarchical approach is also conducive to resource management and optimization, better balancing the loads among different clusters and virtual machines and improving resource utilization.

Adaptability of the Deep Reinforcement Learning Model: The deep reinforcement learning model in this research, through a meticulously designed state space and return function, demonstrates remarkable adaptability to the dynamic changes of the cloud environment. Unlike traditional rule-based or heuristic scheduling algorithms, the deep reinforcement learning model can automatically learn and adapt to the dynamic changes of tasks and resources in the cloud environment. For instance, by modeling the Gaussian distribution of changes in the number of virtual machines, the model can promptly perceive the increase or decrease of virtual machine resources and adjust the task allocation strategy accordingly. When confronted with different load situations, the unique reward function can guide the model to make a reasonable trade-off between task overdue time and cost, thereby achieving satisfactory scheduling performance under various complex load conditions.

Learning and Optimization Capabilities: The deep reinforcement learning model possesses powerful learning and optimization capabilities. Compared with traditional static scheduling algorithms, it can continuously learn in the process of ongoing task scheduling and constantly optimize its own scheduling strategy. Through techniques such as experience replay and target network, the model can effectively utilize historical empirical data for learning, avoid getting trapped in local optimal solutions, and gradually converge to a more optimal scheduling strategy. Such learning and optimization capabilities enable the HITS framework to continuously adapt to the changes of the cloud environment and continuously improve the efficiency and quality of task scheduling during long-term operation.

Comment 5: The methodology section hardly cites any literature for the formulae and state-of-the-art techniques.

Answer: Thanks for good advice. Add the corresponding citations of Reinforcement Learning and Double Deep Q-Network (DDQN) and Playing Atari with Deep Reinforcement Learning. Add four formulae in the methodology section.

30. R. S. Sutton, A. G Barto. Reinforcement Learning: An Introduction. IEEE Transactions on Neural Networks, 1998, 9(5):1054.

31. V. Mnih, K. Kavukcuoglu, D. Silver, A. Graves, I. Antonoglou, D. Wierstra, & M. Riedmiller. Playing Atari with Deep Reinforcement Learning. Computer Science, 2013.

Comment 6: Paper needs to be revised for grammatical errors and typos.

Answer: Thanks for good advice. The grammatical errors and typos of paper has been revised by native English speakers.

Comment 7: Article needs to be proofread from the native English speaker or rewritten in the academic writing.

Answer: Thanks for good advice. The paper has been revised by native English speakers.

Comment 8: The abstract needs to showcase the numerical finding of the research study to reflect the contribution in the field.

Answer: Thanks for good advice. The abstract has been revised.

With the increasing popularity of cloud computing services, their large and dynamic load char-acteristics have rendered task scheduling an NP-complete problem. Aiming at the problem of large-scale task scheduling in cloud computing environment, the paper proposes a novel cloud task scheduling framework using hierarchical deep reinforcement learning (DRL) to address the challenges of large-scale task scheduling in cloud computing. The framework defines a set of virtual machines (VMs) as a VM cluster and employs hierarchical scheduling to allocate tasks first to the cluster and then to individual VMs. The scheduler, designed using DRL, adapts to dynamic changes in the cloud environments by continuously learning and updating network parameters. Experiments demonstrate that it skillfully balances cost and performance. In low-load situations, costs are reduced by using low-cost nodes within the Service Level Agreement (SLA) range; in high-load situations, resource utilization is improved through load balancing. Compared with classical heuristic algorithms, it effectively optimizes load balancing, cost, and overdue time, achieving a 10% overall improvement. Experimental results demonstrate that this approach effectively balances cost and performance, optimizing objectives such as load balance, cost, and overdue time. One potential shortcoming of the proposed hierarchical deep reinforcement learning (DRL) framework for cloud task scheduling could be its complexity and computational overhead. Implementing and maintaining a DRL-based scheduler requires significant computational resources and expertise in machine learning. There are still shortcomings in the method used in this article. Firstly, the continuous learning and updating of network parameters might introduce latency, which could impact real-time task scheduling efficiency. Furthermore, the framework's performance heavily depends on the quality and quantity of training data, which might be challenging to obtain and maintain in a dynamic cloud environment.

Comment 9: The introduction needs to be providing rationale of the study, and brief literature review of existing studies, which is hard to differentiate in its current form.

Answer: Thanks for good advice. The introduction has been revised.

Cloud computing is a resource delivery and usage model. Service providers integrate a large number of nodes into a unified resource pool through virtualization technology, and users obtain the required computing resources through the network [1]. Cloud computing, as one of the core infrastructures in the current field of information technology, faces increasing pressure in task scheduling with the rapid development of big data, the Internet of Things, and 5G technologies. Task scheduling is an important research direction in cloud computing, whose essence is to reasonably allocate user requests to computing nodes for processing. However, this generates a huge solution space, and the optimal solution can-not be obtained in polynomial time, so the task scheduling of cloud computing is an un-certain NP problem [2,3].

Traditional task scheduling methods, whether heuristic algorithms [4] based on simple rules or some metaheuristic algorithms [5], exhibit numerous limitations when dealing with large-scale and dynamically changing cloud tasks [6]. For instance, heuristic algorithms often lack adaptability to complex environmental changes and have difficulty in flexibly adjusting scheduling strategies under different load and resource conditions. Although metaheuristic algorithms can perform global optimization to a certain extent, they have complex parameter settings and high computational overhead, making them difficult to be effectively applied in cloud task scheduling scenarios with high real-time requirements.

Many researchers have studied this problem and proposed heuristic and metaheuristic algorithms to solve it. But the actual cloud computing environment is complicated and dynamic, and traditional methods cope poorly with it. Re-searchers are using reinforcement learning (RL) and deep reinforcement learning (DRL) for learning capabilities to solve the dynamic scheduling problem of cloud computing [7-9]. Due to the diversity of user requests and resources, different quality of service (QoS) constraints must be simultaneously met, and how to respond to large-scale user requests while meeting the requirements of cloud service providers is an urgent problem. Intelligent scheduling algorithms are essential to overcome the difficulties of large-scale task scheduling. In this research, a hierarchical intelligent task scheduling framework (HITS) based on a hierarchical DRL algorithm is proposed. In the scheduling framework, a col-lection of VMs is called a VM cluster. When the framework receives a task request, it allocates the task to a cluster, and then to a VM via the task scheduler inside the cluster. We apply DRL technology to the scheduler, and through the design of the state space and re-turn function of each layer, it can adapt to the dynamic changes of the cloud computing environment, and adjust its scheduling strategy through continuous learning.

Based on these current situations, we propose to adopt the hierarchical deep reinforcement learning technology to address the cloud task scheduling problem. Deep reinforcement learning has powerful learning capabilities and adaptability to complex environments. It can automatically optimize scheduling strategies through continuous interaction and learning with the cloud environment. The hierarchical architecture helps to decompose large-scale problems into manageable sub-problems, improving decision-making efficiency and the system's scalability. We expect that through this innovative method, it is possible to meet the cost control requirements of cloud service providers while providing users with more efficient and reliable services, achieving a comprehensive improvement in multiple aspects such as performance, cost, and flexibility in cloud computing task scheduling, filling the gaps of traditional methods in handling large-scale and dynamic cloud task scheduling, and promoting the further development and application of cloud computing technology in the modern information technology system.

Cost and load balancing are two crucial objectives in cloud task scheduling. From the perspective of cost, cloud service providers need to reduce the cost of resource usage as much as possible to enhance profit margins while meeting user requirements. Different types of virtual machines have diverse cost structures, including computing cost, storage cost, and bandwidth cost. Our scheduler, through the deep reinforcement learning algorithm, comprehensively considers the resource requirements of tasks and the cost characteristics of virtual machines during the task allocation process. For example, when a task arrives, the scheduler evaluates the idle resource situation and the corresponding cost of the virtual machines within each cluster and preferentially assigns the task to the combination of virtual machines or clusters that can meet the task requirements and have a lower cost. Such an approach can effectively reduce the overall cost of task execution and improve the cost-effectiveness of resources.

For load balancing, the motivation is to ensure that the utilization rates of various resource nodes (clusters and virtual machines) in the cloud environment are relatively balanced and avoid situations where some nodes are overloaded while others are idle. This not only helps to improve the overall performance and stability of the system but also extends the service life of hardware devices. In the decision-making process, our scheduler takes the load situations of clusters and virtual machines as important state information and inputs it into the deep reinforcement learning model. By designing a reasonable return function, positive rewards are given to scheduling decisions that can achieve load balancing, and vice versa. For example, when the standard deviation of the virtual machine loads within a cluster is small, indicating a relatively balanced load, the scheduler tends to continue assigning tasks to this cluster. When the load of a certain virtual machine is too high, the scheduler will consider assigning subsequent tasks to other virtual machines or clusters with lighter loads, thereby dynamically adjusting the task allocation strategy to achieve load balancing of resources in the cloud environment and reducing performance bottlenecks and resource waste caused by uneven loads.

---

## [Decision Letter · Decision Letter 1]

21 Jul 2025

An novel cloud task scheduling framework using hierarchical deep reinforcement learning for cloud computing

PONE-D-24-45416R1

Dear Dr. cui,

We’re pleased to inform you that your manuscript has been judged scientifically suitable for publication and will be formally accepted for publication once it meets all outstanding technical requirements.

Kind regards,

Sameena Naaz

Academic Editor

PLOS ONE

Additional Editor Comments (optional):

The manuscript can be accepted for publication

---

## [Editor Report · Acceptance letter]

PONE-D-24-45416R1

PLOS ONE

Dear Dr. Cui,

I'm pleased to inform you that your manuscript has been deemed suitable for publication in PLOS ONE. Congratulations! Your manuscript is now being handed over to our production team.

Kind regards,

on behalf of

Dr. Sameena Naaz

Academic Editor

PLOS ONE